# Analyzing the Spatial Correspondence between Different Date Fruit Cultivars and Farms' Cultivated Areas, Case Study: Al-Ahsa Oasis, Kingdom of Saudi Arabia

**Ahmed I. H. Ismail** [1,2], **Abdalhaleem A. Hassaballa** [3,4,*], **Abdulrahman M. Almadini** [5,†] and **Samah Daffalla** [3]

1   Agri-Business and Consumer Sciences Department, College of Agricultural and Food Sciences, King Faisal University (KFU), Al-Ahsa 31982, Saudi Arabia; aismail@kfu.edu.sa
2   Rural Community & Agricultural Extension Department, College of Agriculture, Ain Shams University, Cairo 11566, Egypt
3   Department of Environment & Agricultural Natural Resources, College of Agricultural and Food Sciences, King Faisal University (KFU), Al-Ahsa 31982, Saudi Arabia; sbalal@kfu.edu.sa
4   Department of Agricultural & Biological Engineering, Faculty of Engineering, University of Khartoum, Khartoum 11111, Sudan
5   Soil Fertility and Environmental Management, Department of Environment & Agricultural Natural Resources, College of Agricultural and Food Sciences, King Faisal University (KFU), Al-Ahsa 31982, Saudi Arabia; almadiniam@gmail.com
*   Correspondence: ahassaballa@kfu.edu.sa; Tel.: +966-549522886
†   Retired.

**Abstract:** Diversity in date palm (DP) cultivars plays a crucial role in the agroecosystems of several countries, such as the Kingdom of Saudi Arabia (KSA). This study aims to map and analyze the spatial distribution of the most grown DP cultivars (Khlas, Ruziz, and Shishi) in the Al-Ahsa oasis in the KSA and to highlight their spatial correlation with the corresponding cultivated patches within farms. Descriptive and spatial data on 288 farms were analyzed using GIS, data curation, cross-TAB statistics, clustering maps, and spatial autocorrelation techniques. The obtained results revealed that most of the oasis's DP farms are within a cultivated area of <500 m$^2$. The larger cultivated areas are mostly in the oasis's northern and central subregions, agreeing with the spatial distribution of trees. In total, 56.9% of the studied farms grew the cultivars together within the least rank (<500 m$^2$) of cultivated area, having the greatest tendency for DP cultivation. Khlas was the most dominant cultivar being the least absent from cultivation with 3.1% compared to Ruziz (31.9%) and Shishi (37.8%). The spatial distribution of DP plantations in the oasis was also consistent with the spatial variation in soils and irrigation water salinity, necessitating the need for special agricultural extension programs. In conclusion, these outcomes indicate that this study is essential for DP sustainability, growers, authorities, and policy makers.

**Keywords:** date palm; cultivars; spatial distribution; sustainability; Al-Ahsa; oasis; KSA

## 1. Introduction

Date palm (*Phoenix dactylifera* L.) cultivation holds a vital segment in the agricultural sector of the Kingdom of Saudi Arabia (KSA) due to its religious, social, and economic values. In the year of 2018, national statistical data indicated that the total number of date palm trees in the country was more than 31 million [1], grown over an area of 207,369.8 ha with a total production of 1,541,769 tons (i.e., ~17% of the world's total production) [2]. There are 400 date palm cultivars grown over 72% of the total permanent cultivated area in KSA [3,4]. Thus, the KSA was globally ranked in the year 2019 among the leading countries that cultivate date palm trees and produce date fruits [5]. These statistical figures are indebted to the generous Saudi government that provides non-profitable and long-term subsides to the growers of date palm trees.

In KSA, the Al-Ahsa (i.e., Al-Hassa) oasis is a prominent agricultural area well known for its intensive date palm cultivation. In the oasis, the estimated total number of date palm trees ranges between 3 and 4 M randomly grown over 70% of the oasis's total cultivated area that covers more than 8200 ha [6]. Numerous studies have stated that there are 40 distinct cultivars in the oasis, where Khlas, Ruziz, and Shishi are the most cultivated and valuable [6–8]. The Al-Ahsa oasis is one of the leading agricultural regions cultivating these cultivars in the KSA according to Elsabea [9], who also stated that these cultivars are well recognized for their distinguished processing features, mainly as soft varieties with diverse colors and pleasant tastes.

Date production in the KSA possesses pronounced economic value. El-Habba and Al-Mulhim [10] indicated that the KSA is ranked among the top nations worldwide in the trade market of dates, yet, external export is low, being estimated as only 6.8% of KSA productions. The authors, however, advocated that the KSA dates have some relative advantages in global markets, considering the availability of high-quality varieties and good processing techniques. Nonetheless, other studies have argued that the quality and age of the date palm cultivars in the KSA offer limiting factors in the investment successes of date palm cultivations [11]. Moreover, Alseleem [12] suggested that date production's economic status and prices are a function of the cultivars, production costs, and farming practices for date palm trees, including soil fertilization, pest control, irrigation, weed removal, pollination etc. Osman and Al-Besher [13] also found that labor costs are the most influential inputs in date palm production, taking into consideration that most of the labor is temporarily hired. Other studies also advised that there are urgent needs to redistribute the economic resources employed in the date palm industry to reduce the production costs and, hence, to increase the capabilities of Saudi dates to compete globally [14].

On the other hand, it is necessary to emphasize that there are certain capacities to strengthen the exports of Saudi dates. Al-Shreed et al. [15], for example, proposed that the Saudi dates have some relative incentives to be more accepted worldwide; these are: providing high-quality products requested by high-income countries, supplying large quantities of low-quality products to meet enquiries from low-income countries, improving exporting packing industries to effectively compete with consumer enquires, coming up with untraditional packed products of dates, and benefiting from the spiritual values of the Saudi dates, particularly within Muslims communities. Mikki [16] also suggested that the Saudi date manufacturers should exercise extra efforts to find new markets, to make advertisements, and to employ propaganda strategies to confront the substantial involved expenses. El-Sebaei and Al-Soliman [8] inferred that, depending on the studied cultivars (i.e., Khlas, Ruziz, and Shishi), the marketing margins of dates are significantly affected by the marketing costs and methods; the technical problems related to the services of transportation, packaging, sorting, marketing, storage of dates; and the behavioral problems related to the knowledge of the marketing methods, good quality, and best pricing techniques.

Few research studies have been conducted in the Al-Ahsa oasis attempting to explore the relationship between the number of date palm trees and the agricultural areas or to determine the density of the trees per cultivated areas. Elprince et al. [17] utilized a BASIC computer program to produce palm density maps and to assess areas growing date palms with dissimilar densities given as percentages of the total cultivated areas. The main finding of this work indicated that the total calculated area was underestimated as compared to the actual area due to using a large-scale map (1:100,000), estimating asymmetrically shaped submain areas, and the random scattering of cultivated areas within the submain areas. Moreover, a recent study conducted by Almadini et al. [6], using a randomly selected sample of 258 active farmers who were surveyed by explicitly designed questionnaires, revealed that the farm features (utilized agricultural area, ownership, and labor) were dissimilar from one region to another throughout the oasis. However, none of these studies used the geographic information system (GIS) technique to assess the correlation between the number of date palm trees and their cultivated areas and, hence, to estimate the spatial

variations in their densities in the oasis. The GIS was successfully used in several studies to depict the topological spatial variations in land cover features [18–22], and in others to fuzzily model such land surface features [23–27].

Therefore, the current study aimed to analyze and map the spatial distribution of the most favorable grown date palm cultivars in the Al-Ahsa oasis, the KSA. The relationships between these cultivars to their corresponding cultivated patches within the farms were also highlighted and discussed. This was carried out to emphasize the current dominating nature of these date palm cultivars at the Al-Ahsa oasis, as a prominent date palm cultivated area in the KSA.

The study is assumed to contribute to the applied date palm research fields through determining the tendency of farmers to exploit the farm's available area based on the preferences of either a particular date palm cultivar or a value of the available cultivated land being converted into non-agricultural purposes. Thus, the outcomes of this investigation can be generalized for other areas. It is also expected to highlight the behavioral disparity of date palm farmers in apportioning the palm trees' area, taking the study area's geographical subregions (northern, central, and eastern) as an example. This could present important economic indicators related to the farm area and the extent to which farmers need to utilize the trees' microenvironment in order to grow other plants. Finally, the study will also contribute to the research stream through emphasizing the importance of the spatial autocorrelation application as a tool to reveal the cultural affinity between each neighboring farmer to choose or cultivate either a specific cultivar or to combine various cultivars, which, in turn, criticizes or supports the superiority of a specific cultivar as the literature states.

In addition, the application of the spatial autocorrelation technique used in this study vigorously characterizes the extent of the relationship between each two variables being able to change within an area. The technique also has the potential to assess the significance of spatial autocorrelation along with any potential spatial dependency between two variables. Thus, such methodology used in the current study could reflect its applicability for any agriproduct (i.e., fruit trees, field crops, and others) being linked to spatial components (i.e., cultivated areas, soil properties, and others).

## 2. Materials and Methods

### 2.1. The Study Area

Al-Ahsa oasis is an area distinguished by date palm cultivation, located in the Eastern province of the KSA about 70 km to the west of the Arabian Gulf coast, 150 km southwest of Dammam port, and 320 km to the east of the capital city Riyadh. It lies between longitude of 49°30′–49°50′ E and latitude 25°20′–25°40′ N (Figure 1), with an elevation of 160 m above sea level in the west that gently inclines toward the Gulf coast [28]. The oasis generally has an "L" shape with an axis extending for 30 km in the south–north direction and the other axis stretching for 18 km in the west–east direction. Both axes intersect in the southwest corner of the oasis at Al-Hofuf city, the largest metropolis in the area [17]. Studies have also pointed out that the oasis is primarily surrounded by active and mobile sand dunes originated from Al-Jafurah desert, which initiates sand encroachments at northern, eastern, and southern boundaries jeopardizing the oasis from the north-west and north routes [18,29].

The climatic data considerably vary between seasons and years in the Al-Ahsa oasis. Based on Mansour [30], the oasis is situated in the sub tropic arid zones, with a climate generally characterized by its very hot–dry summers and cool and relatively dry winters. According to Al-Ali [31], the summer season starts in May to the end of August with a mean temperature ranging from 45 °C to 24 °C, while the winter season begins in November to the end of March with a mean temperature ranging from 21 °C to 8 °C. In summer, the area is considered as the hottest region in the country. The relative humidity varies between 21% and 29% in summer and between 31% and 55% in winter, with a monthly average of 38%. The rainfall is usually in winter and spring seasons with an overall annual mean of 85 mm.

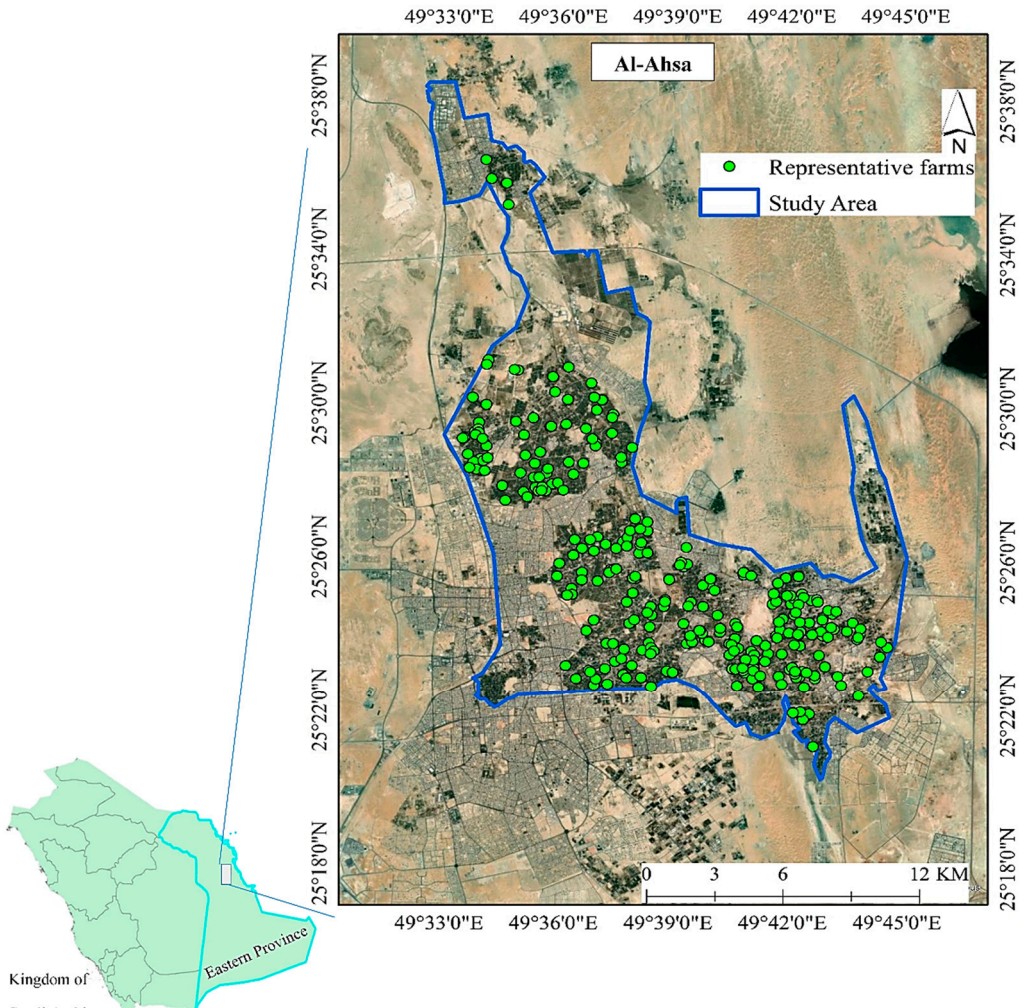

**Figure 1.** Study area map, where Al-Ahsa oasis is delineated in the blue and the distribution of the representative farms is marked in the green.

The total area of the oasis is about 20,000 ha, among which more than 8200 ha are under irrigated cultivation, with 92% being planted with date palm trees [17]. The date trees in the oasis are usually planted at 6 m × 6 m distances, with interspaces being planted with fruits, vegetables, or field crops, depending on the space between the date trees [17,31]. The cultivated area is practically divided into three subregions, namely: the northern (i.e., Jlejilah), the eastern (i.e., Al-Mansourah), and the central (i.e., Al-Harah). The soils of the cultivated lands in the oasis are typified with spatially varied properties that decline their quality and hence productivity. These properties mainly include prevailing light texture, high salinity and calcium carbonate contents, and low contents of organic matter and essential plant nutrients [32]. These soils also are dominated by the six great groups, which are Salorthids, Gypsiorthids, Calciorthids, Psammaquants, Haplargids, and Torripsamments [33].

The main source for irrigation water is groundwater, which is mainly supplied from the karistified neogene aquifer (i.e., 100–180 m deep) that belongs to the Umm-er-Raduma formation [34–36]. Such water is characterized by high (C3) to very high (C4) salinity hazards and low (S1) to medium (S2) sodicity hazards [37,38]. Additionally, water properties were verified by other investigators [39], who also implied that these properties have spatial variations with salinity and its components increasing in the north and north-east directions.

## 2.2. Oasis Statistical Highlights

Agricultural activities in Al-Ahsa oasis are the main source of livelihood for most local people. These activities gained their potential from the available plentiful groundwater in the oasis as indicated earlier. Such water availability gave the oasis its historical and social values as one of the ancient agricultural areas [40]. Historically, the settlements in the oasis go back more than 5000 years, being considered as one of oldest human communities. Nonetheless, recent survey indicated that the population of the region was nearly 1.2 million in 2016, living in 4 major cities and tens of villages, with fewer than 20% of non-citizens [40] and a population growth rate of 3.8% [41]. Due to the date palm production in the area, the Ministry of Environment, Water and Agriculture (MEWA) established a modern date factory that processes its dates' abundant productions [40], which resulted in additional value to the farmers.

The production and processing of Saudi dates attracted special national attention due to the vital status of date cultivation in the country. Between 2015 and 2020, Saudi exports of dates increased 73% in value of returns and 68% in quantity of production, with 107 countries benefiting from these products [42]. In 2020, Saudi date exports equaled 17% of the total national production, most of it with a Saudi date's trademark [42]. The value of these exported dates was nearly 215 million Saudi riyals (i.e., about USD 57.33 million).

## 2.3. Data Collection and Processing

The investigated data of farm's contents and farmer's activities were obtained from the Saudi Irrigation Organization (formerly Hassa Irrigation and Drainage Authority, HIDA), Al-Ahsa, KSA (accessed on 11 November 2020). Throughout the course of data collection, many criteria were taken into consideration, namely: geographical location (i.e., GPS), irrigation water sources (i.e., underground, tertiary treated wastewater, or mixed), farm cultivated areas ($m^2$), irrigation methods (i.e., channel, ditches, pipes, etc.), and crop types. Collected data of farms and farmers were then divided into different categories according to specific criteria throughout the major three subareas over the oasis, which were Jlejilah, Al-Mansourah, and Al-Harah, and then over 25,400 farms were spotted and covered within the farms' network, 6281, 5919, and 3259 farms, respectively. The farms allocated to the subareas were carefully selected to the current study, as they are in line with its objectives.

To set a suitable size for the collected samples that represent all farms with common features all over the oasis, this study applied the proportional stratified random sampling approach constructed by Krejcie and Morgan [43], who belong to the research division of the National Education Association. In this technique, the formula used to determine the sample size of a set of data was as follows

$$S = \frac{X^2 NP(1 - P)}{d^2(N - 1)} + X^2 P(1 - P) \tag{1}$$

where
$S$ = required sample size.
$X^2$ = the table value of chi-square for 1 degree of freedom at the desired confidence level (3.841).
$N$ = the population size.
$P$ = the population proportion being assumed to be 0.50, as it would provide the maximum sample size.
$d$ = the degree of accuracy expressed as a proportion (0.05).

Making use of the proportional stratified random sampling approach [43], the collected number of farms was downsized into 288 representative farms. The applied criteria included difference in oasis section's orientation (i.e., Jlejilah, Al-Mansourah, and Al-Harah), difference in cultivated areas (i.e., small, medium, and large), differences in type of irrigation and irrigation source, and farm's agricultural products (i.e., types, cultivars variety, and quantity).

It is worth stating that there have been more than 40 different cultivars of date fruit observed from the collected data from the Al-Ahsa farms. Hence, the 3 most preferably cultivated cultivars of date fruits were selected by nominating the most frequently cultivated ones in the oasis. Thus, the applied frequency analysis of the collected 25,400 farms' data revealed that "Khlas", "Ruziz", and "Shishi" cultivars were the most cultivated in the oasis (in terms of number of trees/farm), with frequency percentages of 65%, 16%, and 4%, respectively. These 3 cultivars were, hence, applied to depict and map the density (i.e., number of trees per farm) and spatial distributions of date palm trees in the Al-Ahsa oasis.

Cultivar's specific area and distribution were also delineated by identifying farms with an entirely growing sole cultivar, then dividing the farm cultivated area by the number of sole cultivar trees to produce the area of a single tree in m$^2$. Hence, for the farms that had the same criteria, the average total area of the specified cultivar was then calculated. Finally, the spatial distribution of cultivated areas, total palm, and selected cultivars of the representative farms was mapped utilizing a spatial interpolation (inverse distance weight-IDW) technique in the ArcGIS software program's environment.

### 2.4. Statistical Representation of the Distributed Data

A sort of categorical variables correlation was conducted between the farm's cultivated areas and farm's date cultivars that were previously nominated. This was conducted in terms of number of trees and areas cultivated by the nominated cultivars. Cross tabulation (cross-TAB) correlation tool in the statistical package for social sciences (SPSS) software program was utilized. This tool is mostly applied to study the existence/strength of any association between non-continuous variables. Pearson and Spearman correlations were utilized to assess the significance of the statistical relationships between the correlated variables. Pearson correlation measures the linear relationship between two continuous variables, while Spearman correlation examines the monotonic relationship, as its coefficient is based on the ranked values for every variable instead of the natural data [44].

### 2.5. Spatial Data Analysis

A spatial autocorrelation approach has been applied to study the spatial correspondences among farms' cultivated area and the total number of palms, in addition to the number of the three selected cultivars (i.e., Khlas, Ruziz, and Shishi), given as individual variables. The Bivariate Local Indicator of Spatial Autocorrelation (BiLISA) technique was utilized as a method to identify the spatial correlation and the clustering, as well as the significance of the farms' land utilization by growing a specific cultivar of dates. In addition, it was used to examine the agreement in spatial distribution between the cultivated area and the date palm across the oasis sectors. Furthermore, it is worth stating that BiLISA exhibits the extent to which the attributes of the relationship between each two variables can change within the study area.

Moran's index (*I*) was also examined to determine the effectiveness of the spatial autocorrelation as well as to assess the spatial dependency between two obtained variables [45]. The ultimate finding of BiLISA was typically depicted as a map that can help identify the features of the drawn spatial autocorrelations, while Moran's *I* was graphical plot categorized into four groups. Two groups present the positive spatial clusters (i.e., high–high and low–low) that correspond to values actually surrounded by neighboring pixels having similar values. The opposite two groups entail the spatial outliers (high–low and low–high), which mostly agree with values whose neighboring pixels include diverse values. In this study, GeoDa software package [46], that supplies an exceptionally user-friendly environment to utilize spatial data autocorrelations and assessment, was used.

### 2.6. Flowchart of the Process

Figure 2 shows the flow chart of the tasks executed for the spatial correlation analysis of the obtained farms' activity data over the selected 288 representative farms at the Al-Ahsa agricultural oasis. The conducted work encompassed three stages; the first stage

(data collection and preprocessing) included quantitative and qualitative data collection. The quantitative data represented the number of farms that were initially nominated (25,400 farms), and for each farm, data related to the available assets required for the study work were collected (i.e., number of palm trees, total areas of farms, cultivated areas). On the other hand, the qualitative data entailed the types of date fruit cultivars. Thereafter, due to the high density of the collected data, representative farms were selected. The second stage (spatial autocorrelation) entailed a spatial correlation between the different ranks of palm trees, palm trees' sole area, and farms' cultivated areas, which was achieved using BiLISA spatial analysis software program. The third stage (statistical data representation) entailed a statistical analysis carried out between palm tree sole area and the total cultivated area in each farm, in which cross-TAB function of SPSS software program was used.

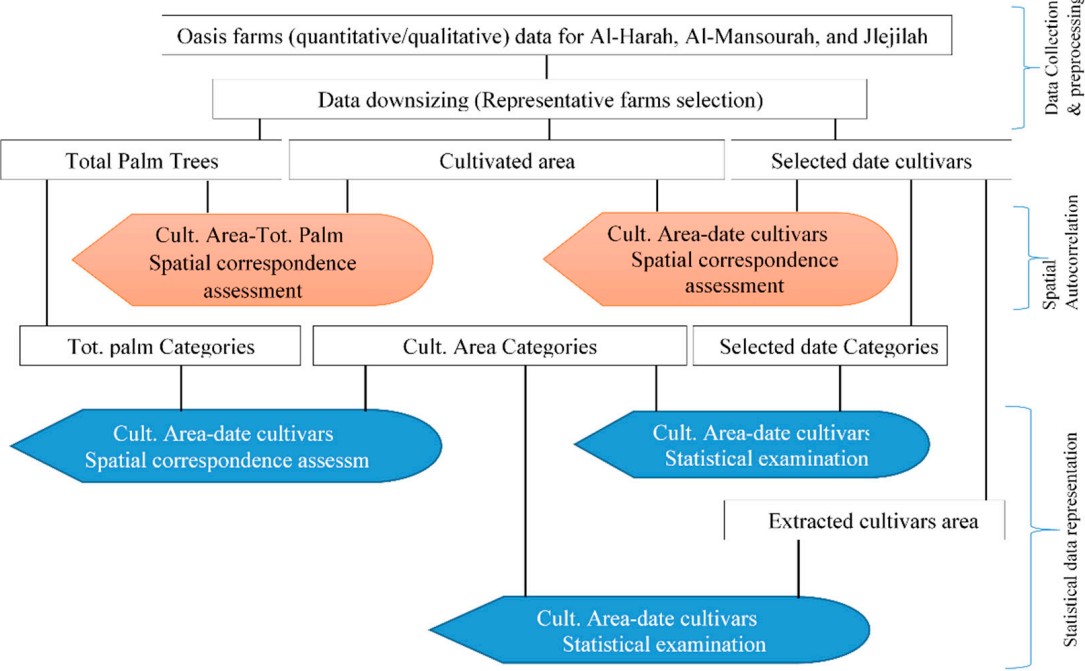

**Figure 2.** Flow chart of the applied spatial and statistical correlations.

## 3. Results

### 3.1. Generated Maps

Based on the extracted 288 representative farms in Al-Ahsa oasis, the quantitative distribution of date palm trees has been mapped using the number of trees per farm, which was categorized into four ranks (i.e., low, <50; medium, 51–100; high 101–200; very high, >200 tree/farm) (Figure 3). The Figure includes four maps describing the spatial distributions of the total palm trees (a), Khlas cultivar trees (b), Ruziz cultivar trees (c), and Shishi cultivar trees (d). It can be noticed from the total palm trees (Figure 3a) that the general orientation of tree density (i.e., 101–200 and >200 tree/farm) is localized in the northern (i.e., Jlejilah) and central (i.e., Al-Harah) subregions of the oasis. However, the eastern subregion (i.e., Al-Mansourah) shows a lower density of trees, as most farms contain 100 trees/farm or fewer (i.e., 50–100 and <50 tree/farm).

The spatial distribution of studied cultivars, however, exhibits that the Khlas cultivar also shows a similar quantitatively dominated and intensive cultivation (i.e., 101–200 and >200 tree/farm) in the northern and central subregions of the oasis, with a much lower tendency in the eastern subregion (Figure 3b). However, the Ruziz cultivar shows a noticeable quantitative concentration of farms in the central subregion (Figure 3c), particularly in the southwestern corner where Al-Hofuf city is located, with a scattered distribution of a few farms in the eastern subregion. The Shishi cultivar, however, displays a limited quantitative intensity of farms in the central subregion as they are mostly localized by the top border

of the cultivated area (Figure 3d). Overall, Figure 3b–d also embodies the distribution of small ranks (i.e., 51–100 and <50 tree/farm) of these cultivars in the oasis, implying their cultivation importance among the farmers of the Al-Ahsa oasis.

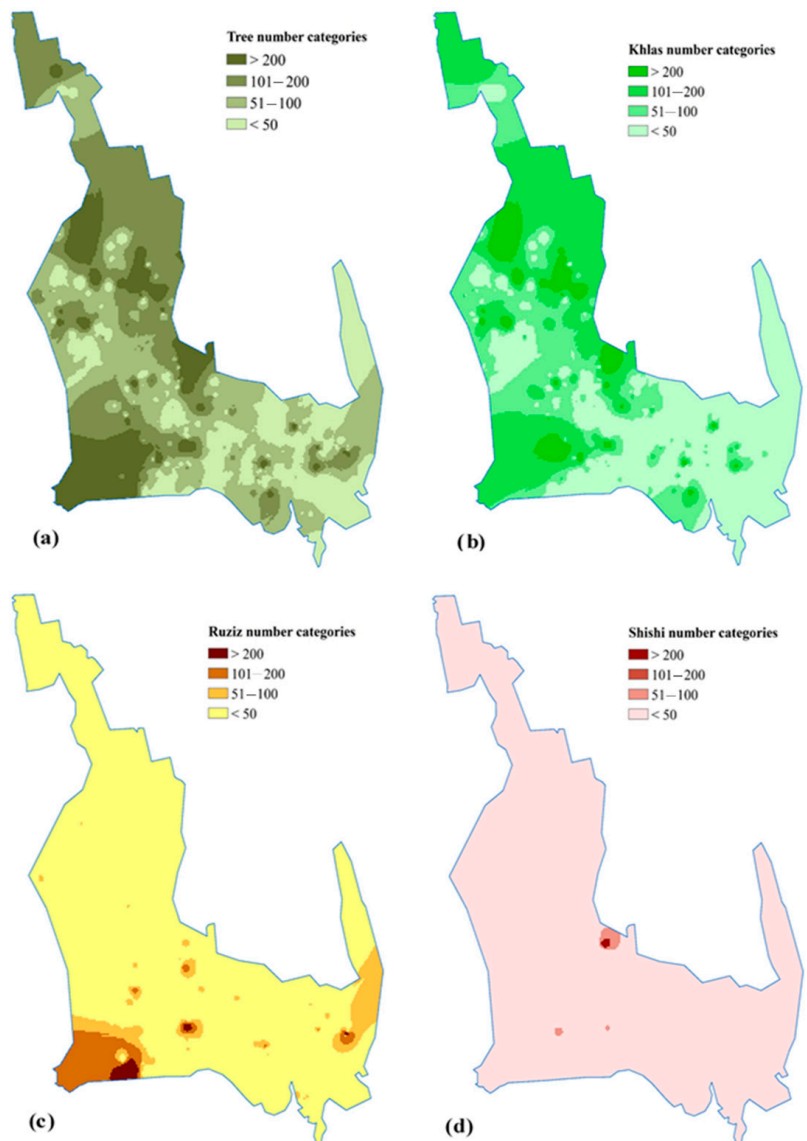

**Figure 3.** The plotted number of palm trees/farm, as in (**a**) the oasis total palm and in cultivars of (**b**) Khlas, (**c**) Ruziz, and (**d**) Shishi.

As indicated earlier, the study intended to plot the spatial distribution of farms' cultivated areas to reveal the differences in the ownership sizes of the farms throughout the oasis. Thus, Figure 4 shows the four maps that depict the distribution of oasis cultivated areas with all cultivars (Figure 4a), the Khlas (Figure 4b), the Ruziz (Figure 4c), and the Shishi (Figure 4d) cultivar. All maps were plotted in four ranks being divided on the basis of the common range of cultivated areas (i.e., <500; 501–1000; 1001–10,000; >10,000 m$^2$). These ranks of cultivated areas were presented as low, medium, high, and very high, respectively, for all cultivars as well as for each sole cultivar.

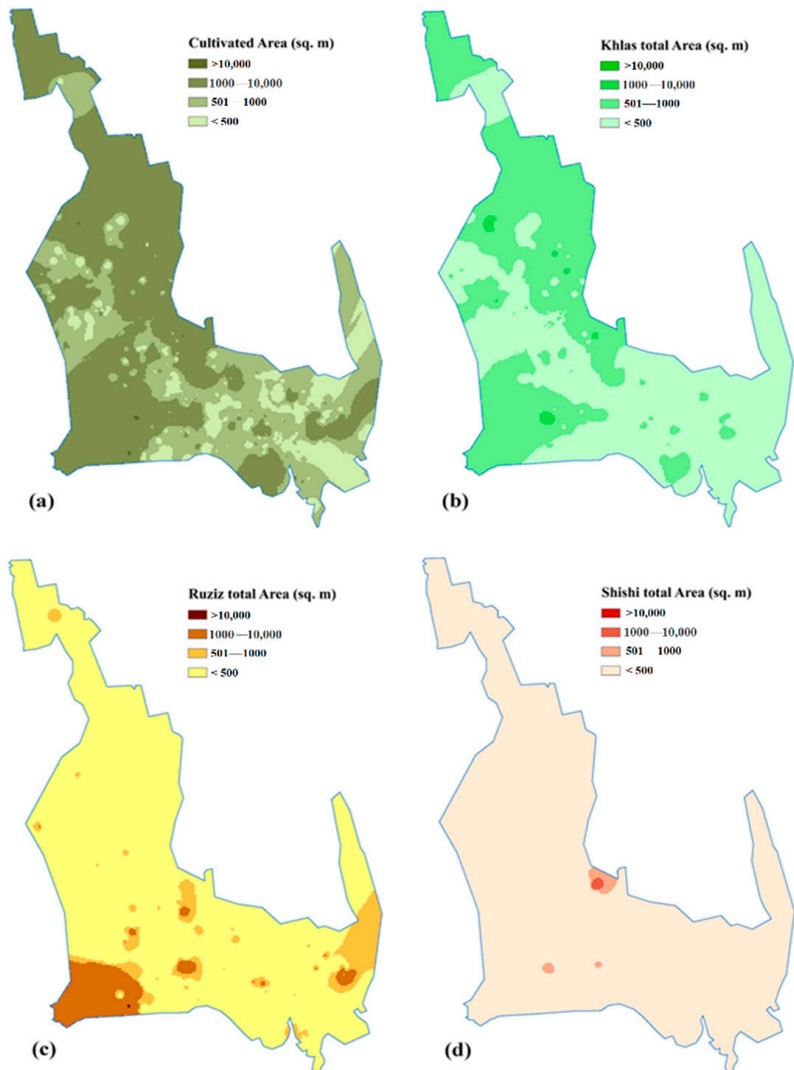

**Figure 4.** The plot of the cultivated area/farm, as in (**a**) the oasis total palm and in cultivars of (**b**) Khlas, (**c**) Ruziz, and (**d**) Shishi.

For all studied cultivars, it is possible to see from Figure 4a that the dominant cultivated area rank was the high class (i.e., 1001–10,000 m$^2$) being sited over most of the Al-Ahsa oasis, followed by the medium class (i.e., 501–1000 m$^2$), with a few small clusters of cultivated areas of very high class (i.e., >10,000 m$^2$) that were depicted in the central subregions of the oasis. On the other hand, the low class farms (i.e., <500 m$^2$) were found mainly in the eastern subregion of the oasis. These findings are in line with the distribution of palm tree numbers in the oasis (Figure 3a). Similarly, the cultivated areas of each cultivar, presented individually in Figure 4b–d for the Khlas, Ruziz, and Shishi, respectively, revealed marked variations in their spatial distribution. The Khlas cultivar, for instance, was found to be the dominant cultivar grown in the oasis, as seen in Figure 4b, which also displays the dense and wide areas of its cultivation throughout the oasis. Its cultivation area was mostly of high (i.e., 1001–10,000 m$^2$) and medium (i.e., 501–1000 m$^2$) classes in the northern and central subregions, with less density in the eastern subregion where low class farms (i.e., <500 m$^2$) were dominant. In contrast, the other two cultivars (i.e., Ruziz, and Shishi) were noticeably found with a prevalent low class all over the oasis (Figure 4c,d, respectively). However, the Ruziz cultivar showed a few sites of high rank mainly in the central and eastern subregions of the oasis (Figure 4c). The observed dominant cultivation of the Khlas cultivar is possibly owed to its economic and social values in the Al-Ahsa oasis and the surrounding regions.

### 3.2. Statistical Representation of the Categorical Data

Table 1 includes the summary of the resultant applied categorical-type correlations between the farms' cultivated area, total number of trees/farm, and palm tree cultivars (i.e., in numbers and cultivated areas). Noticeable patterns of relationships between these traits were perceived in these results. The low rank category (i.e., <50 tree/farm) of the total number of trees embeds most farms growing the three cultivars together (i.e., Khlas, Ruziz, and Shishi), with a total of 164 farms of the surveyed 280 farms (i.e., 56.9%) (Table 1). This low rank is exclusive to the farm cultivated area categories of low (i.e., <500 m$^2$) and medium (i.e., 500–1000 m$^2$) classes. The medium rank category (i.e., 51–100 tree/farm) of the total number of trees included 60 farms (i.e., 20.8%) that were distributed between the medium (i.e., 501–1000 m$^2$) and high (i.e., 1001–10,000 m$^2$) classes of farm cultivated area categories (i.e., 42 and 18 tree/farm, respectively) (Table 1). The categories of high (i.e., 100–200 tree/farm) and very high (i.e., >200 tree/farm) total number of trees included 28 (i.e., 9.7%) and 32 (i.e., 12.5%) farms, respectively. These farms are only found in the categories of high (i.e., 1001–10,000 m$^2$) and very high (i.e., >10,000 m$^2$) classes of farm cultivated areas (Table 1). It is also worth accentuating that the very high category of total trees number is the only rank including farms (i.e., four farms), located in the very high class of the farm cultivated area (Table 1).

**Table 1.** Cross-TAB categorical correlation between cultivated area and palm tree numbers.

| | Categories of Palm Tree Numbers | | Categories of Farm Cultivated Area (m$^2$) | | | | Total |
|---|---|---|---|---|---|---|---|
| | | | <500 | 501–1000 | 1001–10,000 | >10,000 | |
| **Trees' total number** | <50 trees | Count | 142 | 22 | 0 | 0 | 164 |
| | | % within Cult. Area Categ. | 100.0% | 34.4% | 0.0% | 0.0% | 56.9% |
| | 51–100 trees | Count | 0 | 42 | 18 | 0 | 60 |
| | | % within Cult. Area Categ. | 0.0% | 65.6% | 23.1% | 0.0% | 20.8% |
| | 101–200 trees | Count | 0 | 0 | 28 | 0 | 28 |
| | | % within Cult. Area Categ. | 0.0% | 0.0% | 35.9% | 0.0% | 9.7% |
| | >200 trees | Count | 0 | 0 | 32 | 4 | 36 |
| | | % within Cult. Area Categ. | 0.0% | 0.0% | 41.0% | 100.0% | 12.5% |
| **Khlas Cultivar** | No trees | Count | 8 | 0 | 1 | 0 | 9 |
| | | % within Cult. Area Categ. | 5.6% | 0.0% | 1.3% | 0.0% | 3.1% |
| | <50 trees | Count | 134 | 42 | 10 | 1 | 187 |
| | | % within Cult. Area Categ. | 94.4% | 65.6% | 12.8% | 25.0% | 64.9% |
| | 51–100 trees | Count | 0 | 22 | 20 | 0 | 42 |
| | | % within Cult. Area Categ. | 0.0% | 34.4% | 25.6% | 0.0% | 14.6% |
| | 101–200 trees | Count | 0 | 0 | 24 | 0 | 24 |
| | | % within Cult. Area Categ. | 0.0% | 0.0% | 30.8% | 0.0% | 8.3% |
| | >200 trees | Count | 0 | 0 | 23 | 3 | 26 |
| | | % within Cult. Area Categ. | 0.0% | 0.0% | 29.5% | 75.0% | 9.0% |

Table 1. *Cont.*

| Categories of Palm Tree Numbers | | | Categories of Farm Cultivated Area (m²) | | | | Total |
|---|---|---|---|---|---|---|---|
| | | | <500 | 501–1000 | 1001–10,000 | >10,000 | |
| Ruziz Cultivar | No trees | Count | 59 | 18 | 15 | 0 | 92 |
| | | % within Cult. Area Categ. | 41.5% | 28.1% | 19.2% | 0.0% | 31.9% |
| | <50 trees | Count | 83 | 41 | 38 | 0 | 162 |
| | | % within Cult. Area Categ. | 58.5% | 64.1% | 48.7% | 0.0% | 56.3% |
| | 51–100 trees | Count | 0 | 5 | 17 | 3 | 25 |
| | | % within Cult. Area Categ. | 0.0% | 7.8% | 21.8% | 75.0% | 8.7% |
| | 101–200 trees | Count | 0 | 0 | 6 | 0 | 6 |
| | | % within Cult. Area Categ. | 0.0% | 0.0% | 7.7% | 0.0% | 2.1% |
| | >200 trees | Count | 0 | 0 | 2 | 1 | 3 |
| | | % within Cult. Area Categ. | 0.0% | 0.0% | 2.6% | 25.0% | 1.0% |
| Shishi Cultivar | No trees | Count | 78 | 20 | 11 | 0 | 109 |
| | | % within Cult. Area Categ. | 54.9% | 31.3% | 14.1% | 0.0% | 37.8% |
| | <50 trees | Count | 64 | 44 | 66 | 2 | 176 |
| | | % within Cult. Area Categ. | 45.1% | 68.8% | 84.6% | 50.0% | 61.1% |
| | 51–100 trees | Count | 0 | 0 | 1 | 1 | 2 |
| | | % within Cult. Area Categ. | 0.0% | 0.0% | 1.3% | 25.0% | 0.7% |
| | 101–200 trees | Count | 0 | 0 | 0 | 1 | 1 |
| | | % within Cult. Area Categ. | 0.0% | 0.0% | 0.0% | 25.0% | 0.3% |
| | >200 trees | Count | 0 | 0 | 0 | 0 | 0 |
| | | % within Cult. Area Categ. | 0.0% | 0.0% | 0.0% | 0.0% | 0.0% |

From the perspective of the studied cultivars, the results of the cross-TAB correlation revealed that the Khlas cultivar was the most grown in the Al-Ahsa oasis (Table 1), as it was only absent in 3.1% farms (i.e., 9 of 288 surveyed farms), specifically from those of low (i.e., <500 m², with 8 farms, 5.6%) and high (i.e., 1001–10,000 m², with only 1 farm, 1.3%) classes of the farm cultivated area categories. This cultivar however was mostly found in the categorical low rank of total number of trees (i.e., <50 tree/farm) with 64.9% (Table 1), being distributed between the cultivated areas of low and medium (i.e., <500 and 501–1000 m²) classes with 94.4% and 65.6%, respectively. Data also showed that the categorical classes of high (i.e., 1001–10,000 m²) and very high (i.e., >10,000 m²) cultivated areas were the only farms growing Khlas cultivar with ≥101 tree/farm (Table 1).

On the other hand, the other two cultivars (i.e., Ruziz and Shishi) disclosed fewer variations in the categorical ranks of farm cultivated areas and classes of total number of trees per farm than the Khlas cultivar, as seen from the data presented in Table 1, which also point out that the most frequent category of farms with cultivated palms (i.e., number of tree/farm) was the <50 tree/farm, with 64.9%, 56.3%, and 61.1% for Khlas, Ruziz, and Shishi, respectively. Conversely, the least frequent category was the >200 tree/farm, particularly for the Ruziz and Shishi cultivars. These findings are consistent with the nature of the prevalent farms with small areas in the oasis, as already shown in Figure 4 (Table 1).

In addition, Pearson's R (interval by interval) and Spearman correlation (ordinal by ordinal) tests were applied to evaluate both goodness-of-fit and/or the independence of the correlated variables. Their obtained results are outlined in Table 2, which shows that there are distinct significant relationships among the distribution of cultivated area and the distribution of other categorical parameters (i.e., total palm trees and trees of cultivars), with a Pearson's R of 0.88, 0.75, 0.43, and 0.41 for the number of trees of the total palm,

Khlas, Ruziz, and Shishi cultivar, respectively. On the other hand, Spearman correlation produced values of 0.91, 0.76, 0.37, and 0.39, respectively. In addition, all relationships showed approximate significances (i.e., *p*-values of 0.000). This can also be verified by the significant relationship (i.e., $r^2 = 0.9483$) between the farms' total areas and the corresponding cultivated areas (m$^2$) as presented in Figure 5, which shows that farms of medium-to-large-size (10,000–20,000 and up to 40,000 m$^2$) have the least significant correlation, with dispersed points towards farms' total areas (x-axis) reflecting the unsystematic distribution of date palm trees within these specific-size farms (outliers). This is probably due to the common planting methods of the palm trees within these areas having arising business streams. These findings, however, suggest the effective and substantial associations between the cultivated area with the number of grown date palm trees in the study area, insinuating the value of the studied cultivars' cultivation in the Al-Ahsa oasis.

**Table 2.** Pearson's R (interval by interval) and Spearman correlation (ordinal by ordinal) tests.

| Tests | | Categories | | | |
|---|---|---|---|---|---|
| | | Total Palm | Khlas | Ruziz | Shishi |
| **Interval by Interval** | **Pearson's R** | 0.883 | 0.751 | 0.431 | 0.409 |
| **Ordinal by Ordinal** | **Spearman Correlation** | 0.912 | 0.761 | 0.372 | 0.387 |
| **Approximate Significance** | | 0.000 a | 0.000 a | 0.000 a | 0.000 a |
| **N of Valid Cases** | | 288 | 288 | 288 | 288 |

a. Based on normal approximation.

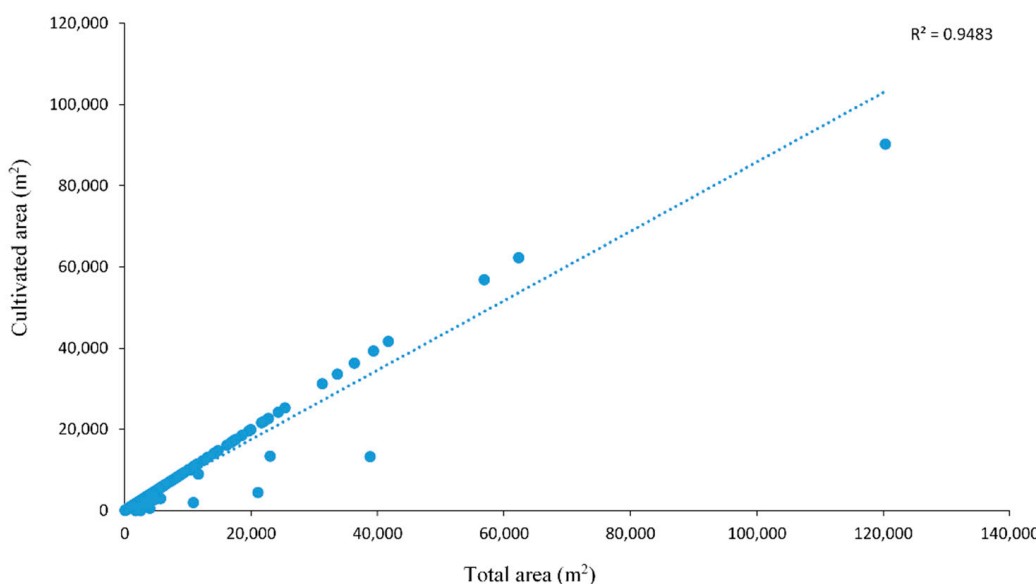

**Figure 5.** A comparative plot of farms' total areas and the corresponding cultivated areas (m$^2$).

The results of the categorical correlation between the farms' cultivated areas and the areas dominated by the studied sole farm's cultivars in the Al-Ahsa oasis are outlined in Table 3. The data in the table reveal that the Khlas was the dominant cultivar in almost all categories of cultivated areas by a sole cultivar. It was the cultivar absent the least in cultivated farms, with only 3.1% of the 288 investigated farms, as compared with the 31.9% and 37.8% for Ruziz and Shishi cultivars, respectively (Table 3). This agrees with the previous suggestion signifying the prevalence of the Khlas cultivar in the oasis as indicated in Figures 3 and 4 and Table 1. In addition, data in Table 3 disclose that the small area of <500 m$^2$ was the prime one cultivated by each of the three studied cultivars with 59.4%, 54.2%, and 60.4% for the Khlas, Ruziz, and Shishi, respectively. On the other hand, according to the categories of farms' cultivated areas (m$^2$) presented in Table 3 and

after excluding the no area category, it is worth highlighting that the cultivated small area retained 134, 83, and 64 farms solely cultivated with the Khlas, Ruziz and Shishi cultivars, respectively. Thus, this emphasizes the tendency of the small areas for cultivation in the Al-Ahsa oasis.

**Table 3.** The categorical correlation between total areas cultivated by sole cultivars and farms' cultivated areas.

| Categories of Areas Cultivated by Sole Cultivar | | | Categories of Farms Cultivated Area (m$^2$) | | | | Total |
|---|---|---|---|---|---|---|---|
| | | | <500 | 501–1000 | 1001–10,000 | >10,000 | |
| **Khlas Cultivar** | **No Area** | Count | 8 | 0 | 1 | 0 | 9 |
| | | % within Cultivated Area Category | 5.6% | 0.0% | 1.3% | 0.0% | 3.1% |
| | **<500 m$^2$** | Count | 134 | 28 | 8 | 1 | 171 |
| | | % within Cultivated Area Category | 94.4% | 43.8% | 10.3% | 25.0% | 59.4% |
| | **501–1000 m$^2$** | Count | 0 | 36 | 14 | 0 | 50 |
| | | % within Cultivated Area Category | 0.0% | 56.3% | 17.9% | 0.0% | 17.4% |
| | **1001–10,000 m$^2$** | Count | 0 | 0 | 55 | 2 | 57 |
| | | % within Cultivated Area Category | 0.0% | 0.0% | 70.5% | 50.0% | 19.8% |
| | **>10,000 m$^2$** | Count | 0 | 0 | 0 | 1 | 1 |
| | | % within Cultivated Area Category | 0.0% | 0.0% | 0.0% | 25.0% | 0.3% |
| **Ruziz Cultivar** | **No Area** | Count | 59 | 18 | 15 | 0 | 92 |
| | | % within Cultivated Area Category | 41.5% | 28.1% | 19.2% | 0.0% | 31.9% |
| | **<500 m$^2$** | Count | 83 | 38 | 35 | 0 | 156 |
| | | % within Cultivated Area Category | 58.5% | 59.4% | 44.9% | 0.0% | 54.2% |
| | **501–1000 m$^2$** | Count | 0 | 8 | 15 | 3 | 26 |
| | | % within Cultivated Area Category | 0.0% | 12.5% | 19.2% | 75.0% | 9.0% |
| | **1001–10,000 m$^2$** | Count | 0 | 0 | 13 | 0 | 13 |
| | | % within Cultivated Area Category | 0.0% | 0.0% | 16.7% | 0.0% | 4.5% |
| | **>10,000 m$^2$** | Count | 0 | 0 | 0 | 1 | 1 |
| | | % within Cultivated Area Category | 0.0% | 0.0% | 0.0% | 25.0% | 0.3% |
| **Shishi Cultivar** | **No Area** | Count | 78 | 20 | 11 | 0 | 109 |
| | | % within Cultivated Area Category | 54.9% | 31.3% | 14.1% | 0.0% | 37.8% |
| | **<500 m$^2$** | Count | 64 | 44 | 65 | 1 | 174 |
| | | % within Cultivated Area Category | 45.1% | 68.8% | 83.3% | 25.0% | 60.4% |
| | **501–1000 m$^2$** | Count | 0 | 0 | 2 | 2 | 4 |
| | | % within Cultivated Area Category | 0.0% | 0.0% | 2.6% | 50.0% | 1.4% |
| | **1001–10,000 m$^2$** | Count | 0 | 0 | 0 | 1 | 1 |
| | | % within Cultivated Area Category | 0.0% | 0.0% | 0.0% | 25.0% | 0.3% |
| | **>10,000 m$^2$** | Count | 0 | 0 | 0 | 0 | 0 |
| | | % within Cultivated Area Category | 0.0% | 0.0% | 0.0% | 0.0% | 0.0% |

Furthermore, the assessment of goodness-of-fit and the independence of the correlated variables were conducted by applying the Pearson's R (Interval by Interval) and Spearman correlation (Ordinal by Ordinal) tests. The results of these tests are summarized in Table 4, which shows that there was a high level of significance between the correlated variables. The correlation between the distributions of farms' cultivated area and the distributions of Khlas, Ruziz, and Shishi cultivars' cultivated areas produced Pearson's R values of 0.81, 0.45, and 0.42, respectively. However, all obtained relationships revealed approximate significances of *p*-values of 0.000 (Table 4).

**Table 4.** Pearson's R (Interval by Interval) and Spearman (Ordinal by Ordinal) tests for cultivated area versus cultivars' area correlations.

| **Tests** | | **Categories** | | |
|---|---|---|---|---|
| | | **Khlas** | **Ruziz** | **Shishi** |
| **Interval by Interval** | **Pearson' s R** | 0.810 | 0.453 | 0.424 |
| **Ordinal by Ordinal** | **Spearman Correlation** | 0.806 | 0.388 | 0.397 |
| **Approximate Significance** | | 0.000 a | 0.000 a | 0.000 a |
| **N of Valid Cases** | | 288 | 288 | 288 |

a. Based on normal approximation.

### 3.3. Spatial Representation of Data Distribution

The approaches of both Bivariate LISA (i.e., BiLISA) and Moran's *I* were employed to conceive any potential correspondence in the variation patterns from the spatially distributed farms in terms of cultivated areas against the spatial distribution of the total number of palm trees, as well as the spatial distribution of the selected date cultivars. The BiLISA test was applied on the obtained parameters, while the Moran's *I* test was used to examine just two parameters (i.e., farms' cultivated areas and total number of palm trees), in which the vertical axis indicates the neighboring values of farms' cultivated areas, and the horizontal axis indicates the total number of palm trees/farm, given as farms' average values. The BiLISA clustering map presented in Figure 6a discloses the distinguished significance, particularly between the oasis' total cultivated area and the total date palm trees. The farms in the northern part of the oasis, however, showed noticeable spatial correspondence (Figure 6b), specifically between the high number of palm trees and the bigger cultivated areas, with a high significance rate (*p*-value 0.001). In addition, the obtained bivariate Moran's *I* was assessed to be 0.16 with 999 permutations and 0.001 pseudo *p*-value (Figure 6c). These results verify the strong relationships between the farms' cultivated area and the number of trees per farm, which is consistent with the previously obtained outcomes of this investigation.

In addition, the BiLISA clustering maps between the oasis cultivated area and the selected date cultivars show a significant correlation between the spatial distribution of cultivated areas and the number of sole cultivar trees (Figure 7). Thus, a noticeable number of farms of the Khlas cultivar in the northern subregion showed a relatively high agreement between the high number of cultivar trees with the bigger assigned cultivated areas (Figure 7a). The same status applies for the Shishi cultivar in the northern subregion (high–high class); yet the most significant correspondence observed between the low trees number and the smaller cultivated areas for the studied cultivars (i.e., Khlas, Ruziz, and Shishi) was in the eastern subregion of the study area (low–low class). This implies the plausible connections between the cultivated areas with the number of trees planted in these areas in the oasis.

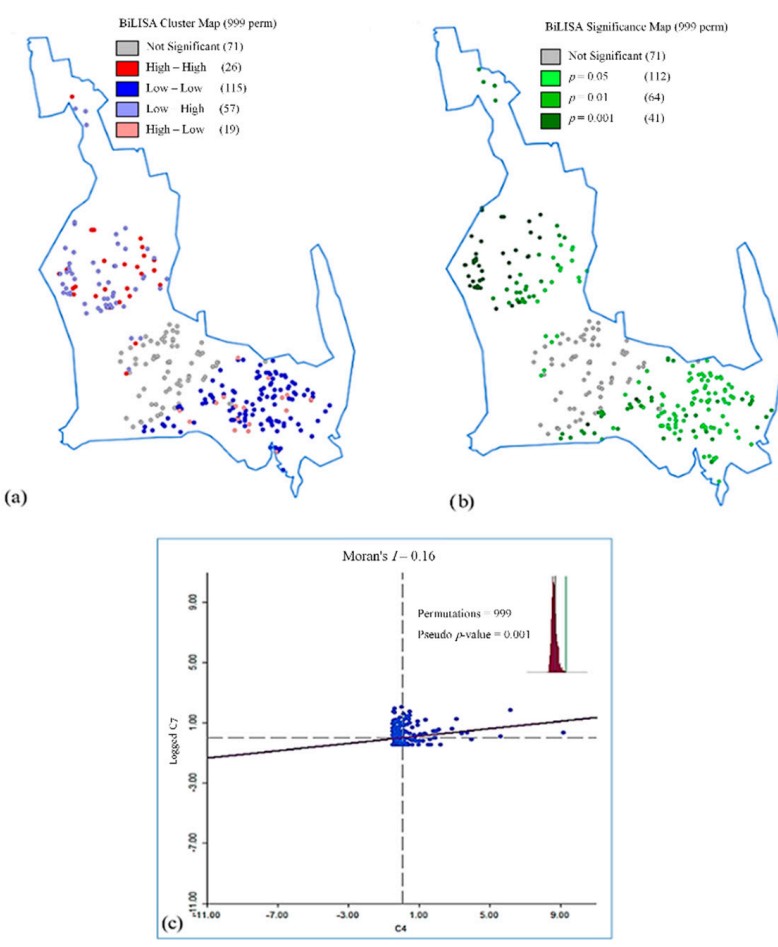

**Figure 6.** BiLISA (**a**) clustering and (**b**) significance maps, for total date trees number plotted against total cultivated area, examined by (**c**) Moran's *I*.

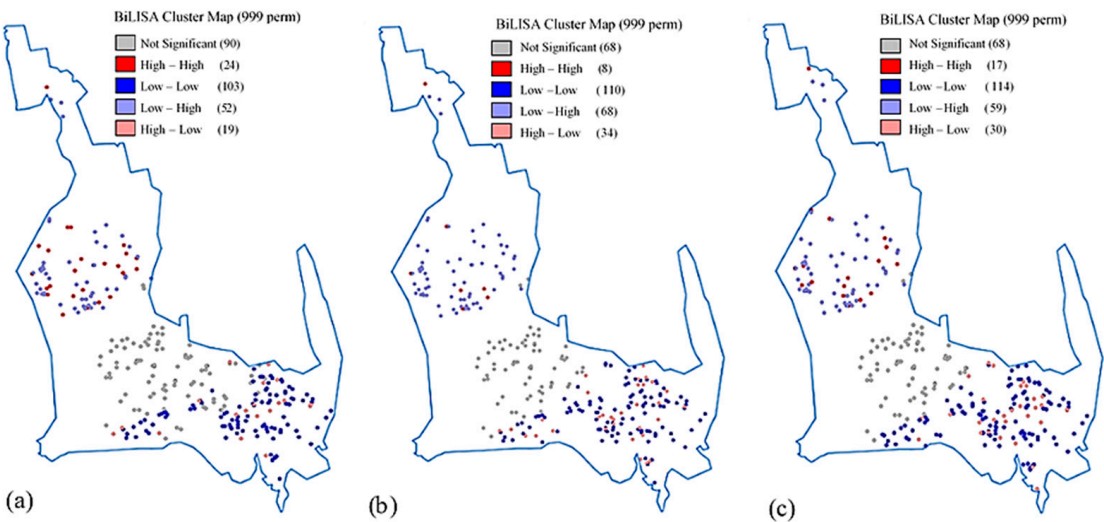

**Figure 7.** BiLISA clustering maps showing the spatial correspondence between cultivated areas and (**a**) Khlas, (**b**) Ruziz, and (**c**) Shishi cultivars as the number of trees.

## 4. Discussion

The obtained findings of this study indicate that the density of total date palm trees (i.e., number of tree/farm) in the Al-Ahsa oasis is mainly oriented to the northern (i.e., Jlejilah) and central (i.e., Al-Harah) subregions of the oasis with tendencies of high and

very high (i.e., 101–200 and >200 trees/farm, respectively) ranks (Figure 3a). Yet, the lowest density of trees in a farm is in the eastern subregion (Al-Mansourah), where most farms contain <50 tree/farm (Figure 3a). These findings are in line with the spatial distribution of cultivated areas, where the high (i.e., 1001–10,000 m$^2$) and very high (i.e., >10,000 m$^2$) classes also exist in the former two subregions (Figure 4a). Moreover, such outcomes can be ascertained by the results of the applied categorical-type correlations between the farms' areas, the total numbers of palm trees/farm, and the palm tree cultivars (i.e., numbers and cultivated areas) (Tables 1–4). On the other hand, the small agricultural holdings (i.e., cultivated area <500 m$^2$) that lay at the central subregion of the oasis have shown significant correspondence (Figure 7a,b) as the small farms' areas are entirely utilized by palm trees.

Thus, these findings may be justified by the dominant small sizes of the agricultural farms in the Al-Ahsa oasis [17,34]. Elprince et al. [17], for instance, indicated that 92% of the total agricultural land in the oasis is cultivated with date palm trees that are planted at a spacing of 6.20 m × 6.20 m in the east and 7.46 m × 7.46 m in the north with the interspacing being planted with some fruit trees and other crops. In addition, Shibah [47] reported that almost 80% of farms in the oasis have sizes between 0.5 and 4 ha (i.e., 1 to 10 acres). Such small farms mainly resulted from physical and sociological factors [34,48,49]. The physical factors are related to the shortage of groundwater to meet growing demands for irrigation water over times; the land's topographical nature limiting the size and distribution of farms; and the main climatic data featured by evapotranspiration high rates and periodic strong dry winds. The sociological factors are linked with the inheriting Islamic law that allows lands to be divided between inheritors leading eventually to smaller farms. The latter factor, however, is well observed in the old-cultivated areas available in the central and eastern subregions.

Moreover, the current study denoted the cultivation superiority of the Khlas cultivar in the Al-Ahsa oasis over the other studied cultivars of Ruziz and Shishi, whether in the spatial distribution of the number of trees/farm (Figure 3b) or in cultivated areas (Figure 4b). The obtained data revealed that the Khlas cultivar was the cultivar absent the least from the studied farms (i.e., 3.1%, 9 of 288 farms,), as compared with the Ruziz cultivar (i.e., 31.9%) and the Shishi cultivar (i.e., 37.8%) (Tables 1 and 3). This domination of the Khlas cultivar was also verified by the results of the cross-TAB correlation that showed it as being mostly grown in the low rank of the total number of trees (i.e., <50 trees/farm) with 64.9%, distributed between the cultivated areas of low class (i.e., <500) and medium class (i.e., 501–1000 m$^2$), with 94.4% and 65.6%, respectively (Table 1). The data also confirmed that this cultivar is the only one found in the categorical classes of high (i.e., 1001–10,000 m$^2$) and very high (i.e., >10,000 m$^2$) cultivated areas. These findings of the Khlas superiority in the oasis have also been concluded by Almadini et al. [6], who suggested that the Khlas cultivation is the most common grown cultivar with 67.15%, followed by the Ruziz cultivar (i.e., 18.87%) and the Shishi cultivar (i.e., 6.96%). Asif et al. [50] also stated that the Khlas cultivar is widely planted in the Al-Ahsa oasis and ranked by many people as the best date cultivar in the world, as it is a fancy date making a delicacy as both a fresh and dry fruit that cures well and retains its flavor even in storage. The dominance of the Khlas cultivar might be explained by its economic and social values in the Al-Ahsa and the surrounding communities as indicated by other investigators [9,11]. In fact, and according to the Saudi General Authority for Statistics, the Khlas cultivar is the most planted date palm tree in the Kingdom, with 25% out of the total 31,234,155 trees [1].

In addition, the obtained results also show marked significant relationships between the distribution of cultivated land and the distribution of the total palm trees, as well as between the spatial distribution of the cultivated areas and the distribution of the number of the sole cultivar trees (Tables 2 and 4 and Figures 5–7). Moreover, the BiLISA clustering map proved that there were significant relationships between the oasis' total cultivated area and the total date palm trees (Figure 6a). The northern subregion farms showed distinct spatial correlations (Figure 6b), particularly between the high number of palm trees and the bigger cultivated areas with a high significance value (*p*-value 0.001). These correlations

evidently sustain the hitherto observed implications in this study that advocate the effective and substantial relationships between the cultivated areas with the planted date palm trees in the Al-Ahsa oasis, whither in total or per farm. This may plausibly assume the common planting methods of the date palm trees in the oasis. The various impacts of the variations in soil properties and irrigation water quality on the spatial distributions of the date palm plantations in the oasis can also be deduced from these relationships. Elevated soil salinity in the oasis is a prime obstacle to its sustainable farming activities [32,48]. In addition, various studies have shown that irrigation water in the oasis is recognized by its high salinity hazards due to its excessive salinity contents [39]. This salinity of irrigation water, however, tends to increase in the direction from central to eastern subregions [39], being consistent with the spatial variations in soil salinity [19,20]. This direction, however, is also in harmony with the observed spatial distributions of date palm plantations in the oasis.

Though date palm trees are salinity tolerant [51,52], several studies have elucidated that increased salinity levels of soils and irrigation water conceivably induce undesirable impacts on their growth traits, nutrient contents, and biomass products [51,53]. Various studies have also revealed that date palm varieties differ in their salinity tolerance [54–56], with some asserting that the Khlas cultivar is more tolerant than others [57–59]. Thus, it is feasible to conclude from these findings that date palm plantations in the oasis demand special farming programs that consider the escalated salinity of soils and irrigation water. These programs ought to seek to improve the production of the date palm and attain the agricultural sustainability of the oasis, ultimately meeting the goals of the Saudi 2030 strategic vision (Unified National Platform, 2021). These goals involve, in part, attaining food security, preserving natural and environmental resources, sustaining the agriculture sector, and improving public income.

**5. Conclusions**

The current study, unprecedentedly, has elucidated some insights on the spatial distributions of the date palm plantations of the most important cultivars (i.e., Khlas, Ruziz, and Shishi) in the Al-Ahsa oasis, as a prime area of date palm cultivation in the KSA. The applied techniques of GIS, and statistical and spatial methods have shown that most of the oasis farms lay within the cultivated area rank of <500 m$^2$, followed by a relatively balanced number of farms within the area ranks of 501–1000 and 1001–10,000 m$^2$. It has also been observed that the ranked cultivated large areas are mostly in the northern and central subregions of the oasis, which is in line with the distribution of date palm trees, whether in total numbers or per farm. The three cultivars were grown together in 56.9% of the studied farms, mostly in farms within a rank of a <500 m$^2$ cultivated area that had the greatest tendency for date palm cultivation in the oasis, whereas they were not found together in farms with a cultivated large area (i.e., >10,000 m$^2$) that solely had the Khlas cultivar. The Khlas cultivar was the most dominant as it was the least absent from all categories of cultivated areas, with only 3.1% compared with 31.9% for Ruziz and 37.8% for Shishi cultivars. The obtained findings of the statistical correlation from cross-TAB and the BiLISA clustering maps noticeably disclosed distinguished and significance relationships in the oasis between the distribution of cultivated areas and date palm trees.

The applied Pearson's R (Interval by Interval) and Spearman correlation (Ordinal by Ordinal) tests indicated that the correlation values between the distribution of farms' cultivated areas and the distribution of the three cultivars' cultivated areas were, respectively, 0.81, 0.45, and 0.42, with approximate significances of 0.000 *p*-values. Moreover, the obtained bivariate Moran's *I* was 0.16 with 999 permutations and 0.001 pseudo *p*-value. Thus, these results substantiate the strong associations between the farms' cultivated areas and the number of trees per farm or in total. This spatial distribution of the date palm plantations in the oasis was also observed to be consistent with the spatial variation in the salinity aspects of soils and irrigation water, necessitating the need for special agricultural extension programs. These programs ought to take into account the variability in salinity to sustain date palm production and manage the agricultural sustainability of the oasis. It may

also be conceived that the outcomes of the present study are precious to the sustainability of the date palm in the Al-Ahsa oasis, and for the growers, the national authorities of agriculture and environment, and policy makers.

Finally, the results of the current investigation suggest the need for further studies that correlate the spatial distribution of the date palm cultivars with other abiotic factors such as: soil properties, irrigation water quality, and agricultural practices. Moreover, other studies related to the impact of agricultural extension services on date palm plantations are necessitated in order to improve farmers' skills aiming to sustain such plantations. Another group of studies may be suggested that deal with modelling the relationship between the cultivated area and the date palm plantation, using some of the new techniques.

**Author Contributions:** Conceptualization, A.I.H.I., A.M.A., A.A.H. and S.D.; methodology, A.M.A. and A.A.H.; software, A.A.H.; validation, A.I.H.I., A.M.A., A.A.H. and S.D.; formal analysis, A.I.H.I., A.A.H. and A.M.A.; investigation, A.I.H.I. and A.M.A.; resources, A.I.H.I.; data curation, A.A.H. and A.M.A.; writing—original draft preparation, A.A.H.; writing—review and editing, A.I.H.I., A.M.A. and S.D.; visualization, A.I.H.I., A.M.A., A.A.H. and S.D.; funding acquisition, A.I.H.I. All authors have read and agreed to the published version of the manuscript.

**Funding:** This work was supported through the Annual Funding track by the Deanship of Scientific Research, Vice Presidency for Graduate Studies and Scientific research, King Faisal University, Saudi Arabia [Project No. AN000371].

**Acknowledgments:** The authors are thankful to the Deanship of Scientific Research, KFU for funding the research and easing the potential of making this work publishable. The authors are also grateful to the Saudi Irrigation Organization, Al-Ahsa, KSA for providing the raw data.

**Conflicts of Interest:** The authors declare no conflict of interest.

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
