# Peer review of "Analyzing the Spatial Correspondence between Different Date Fruit Cultivars and Farms’ Cultivated Areas, Case Study: Al-Ahsa Oasis, Kingdom of Saudi Arabia"

_applsci, doi:10.3390/app12115728_

Round 1

Reviewer 1 Report

This paper aims to analyze the spatial correlation between dates fruit cultivars and cultivated areas using data from KSA. Authors used spatial statistics and Moran’I index to reveal the spatial correlation, and Cross-TAB and Pearson’R statistics were used to detect the relationship between cultivated area and palm tree numbers. This paper is well organized and technically feasible. While, the methods used in this paper are relatively simple, this makes this paper like technical reports. It’s unsure what’s the contribution of this paper, and how this paper differs from other papers. Readers may prefer to see more interesting result. This is my main concern. Specifically, I have the following comments:

>Introduction Part: please address the contribution of this paper.

>BiLISA was used in this paper to reveal the relationship between two parameters: farms cultivated areas and total number of palm trees. It’s too simple that may be useless for fruit cultivars. I suggest to explore more factor that may affect fruit cultivars, such as sunshine duration, soil moisture, and other environmental factors.

>GIS model, such as GWR (Geographically weighted regression) is able to reveal the spatial dependence at different places, that means, factor that influence number of palm trees may differ at different places across the study area.

Reviewer 2 Report

The authors use spatial autocorrelation to determine the best date palm cultivars for cultivation in Kingdom of Saudi Arabia. The paper is well written and of interest to the readership of the journal. I suggest acceptance of the manuscript in its present form.

Reviewer 3 Report

In this article, the authors conduct spatial data analysis on 288 farms in Al-Ahsa Oasis, Kingdom of Saudi Arabia. The analysis is based on well-known methods and shows some interesting correlations, especially in describing the spatial distribution of date palm plantations.

Some positive points of the article:

1 – The context of the case study is relevant

2 – The methods employed in the article can be also used in other similar contexts

3 – It presents a number of analyses and a good discussion aiming to understand them

Some negative points of the article:

1 – The article does not include a good discussion on related work (that is, similar studies)

2 – The article does not present future work topics

General questions and comments:

1The authors should include a Related Work Section that clearly describes similar case studies and explains the ideas borrowed by related work. Further, the authors should discuss how the methods employed in the article differ from those of existing studies.

2 – Figure 2 should be described in the text (Section 2.6).

3 – The generated maps are very interesting and show the coverage area of a set of categories. The authors are encouraged to include a discussion on how such maps relate to the use of fuzzy spatial objects. For instance, in the work “Carniel AC, Galdino F, Philippsen JS, Schneider M (2021). “Handling Fuzzy Spatial Data in R using the fsr Package.” In ACM SIGSPATIAL International Conference on Advances in Geographic Information Systems, 526-535. doi: 10.1145/3474717.3484255.” (and related papers to spatial data science) the authors represent categories of spatial phenomena (e.g., number of trees) by using linguistic terms (e.g., high, medium, low) and allowing that a point belongs to distinct categories with different membership degrees (that denote different intensities). Such a topic can be included as a future work topic since future researchers can improve the analysis conducted in the article by using this kind of approach.

4 – What are the future topics? Please, list the open problems in the area so that other researchers can contribute (the current version is very, very summarized).

Some typos and minor corrections:

“On the other hands” → On the other hand

Please include the thousands separator in all numbers (e.g., line 166, and so on)

Please break the long paragraph of the Conclusion Section into at least other two paragraphs.
